# High-Quality Development of Chinese Agriculture under Factor Misallocation

**DOI:** 10.3390/ijerph19169804

**Published:** 2022-08-09

**Authors:** Shuai Qin, Zheying Han, Hong Chen, Haokun Wang, Cheng Guo

**Affiliations:** 1College of Economics and Management, Northeast Forestry University, Harbin 150040, China; 2Business School, Xuzhou University of Technology, Xuzhou 221018, China

**Keywords:** factor misallocation, high-quality agricultural development, industrial structure upgrade, agricultural science and technology progress

## Abstract

Optimizing factor allocation is the premise of promoting high-quality development of agriculture. Based on the panel data of 31 provinces in China from 2004 to 2020, this paper examines the relationship between factor mismatch and high-quality agricultural development. We found that the high-quality development level of China’s agriculture shows a state of fluctuation and improvement, but the overall level is relatively low and the inter-provincial difference is expanding. Factor mismatch significantly inhibited the improvement of agricultural high-quality development, and the inhibition effect showed obvious temporal and spatial heterogeneity. We also found that the allocation of factors in extreme cases will lead to a 0.01% inter-provincial difference in the high-quality agricultural development. However, with the optimization and upgrading of the agricultural industrial structure and the improvement of the agricultural science and technology, the inhibitory effect of factor mismatch on high-quality agricultural development is constantly weakening. The above conclusion still holds after a series of robustness tests. The conclusions of this paper enrich the theoretical literature on the influencing factors of high-quality agricultural development, and provide an empirical reference for the policy maker of reducing factor mismatch and promoting high-quality agricultural development.

## 1. Introduction

Agriculture is often deemed a national security priority by countries, for it not only provides food for human survival, but also provides a source of employment and rural development [1,2,3]. In particular, the importance of agricultural development to human society has been further highlighted with the continuous spread of COVID-19 and outbreak of armed conflicts in some regions [4,5,6]. China fed about 20% of the world’s population with just 7% of the global agricultural land [7]. However, this outstanding achievement has been partially overshadowed by agriculture-related large-scale environmental pollution [8], which means that the production mode of simply increasing factor input is not sustainable [9].

Given the resource endowment, the rational allocation of production factors is an important way to improve productivity and promote economic high-quality development [10,11]. With the change in residents’ food consumption structure [12], improvement of urbanization level [13], and gradual optimization of agricultural production layout and other external conditions, China’s agriculture is facing a time window from the pursuit of high yield to high quality [14]. Previous studies have proved that optimizing factor allocation is the prerequisite for promoting high-quality agricultural development [15]. However, production factors cannot flow freely and be allocated effectively in accordance with market rules, because regional market segmentation and factor market-oriented reform lag behind [16], leading to the factor mismatch problem in China’s agriculture for a long time [17,18].

In order to improve the efficiency of factor allocation, the Third Plenary Session of the 18th Communist Party of China (CPC) Central Committee proposed to allow the market to play a decisive role in resource allocation, and the 19th CPC National Congress further took market-oriented allocation of factors as one of the key points of economic system reform. After long-term efforts, China’s factor allocation efficiency has been significantly improved, but the factor mismatch problem still exists [19]. Therefore, eliminating the influence of factor mismatch on agricultural production has become one of the effective ways to promote high-quality agricultural development.

There are still some differences and debates on how to evaluate high-quality agricultural development, although the academic community has reached a consensus on its urgency and necessity. Some scholars hold that green total factor productivity not only incorporates environmental factors, but also reflects the technological progress and efficiency improvement in the process of economic development [20], and is a reasonable indicator to measure high-quality agricultural development [21]. For example, Wang et al. (2019) [14], using the SBM–ML index, found that the average annual growth rate of high-quality agricultural development in China was 3.1% from 2003 to 2016. Other scholars’ suggestion that high-quality development is multi-dimensional, and using a single indicator to describe it will lead to a deviation in empirical results [22]. Therefore, total factor productivity is not enough to comprehensively summarize the level of high-quality development [23], and a comprehensive evaluation system should be constructed. For example, guided by the new development concept, Ji (2021) [24], using the AHP–entropy method, found that China’s high-quality agricultural development presents a distribution characteristic of high in the east and low in the west.

By comparison, total factor productivity cannot accurately summarize the rich connotation of high-quality development because of its single index [22], while the comprehensive evaluation method can describe its characteristics relatively comprehensively through the selection of indicators [24]. As the focus of this paper is to grasp the basic characteristics of China’s high-quality agricultural development, the comprehensive evaluation method is taken to measure the agriculture high-quality development. In summary of previous research, this paper holds that high-quality agricultural development is an innovative, coordinated, green, open, and shared dynamic development under the guidance of new development concepts, and is aimed at meeting people’s growing needs for a better life [25].

The influence of factor mismatch on agricultural production has been a hot topic for a long time. In recent years, many literatures have demonstrated the loss of agricultural efficiency caused by factor mismatch, which can be divided into two parts according to the research perspectives.

The first is the single factor mismatch perspective. Scholars have discussed the influence of mismatch of traditional agricultural production factors such as labor [26,27], land [28,29], and capital [30,31], as well as modern production factors such as science and technology [32] and human capital [33], on agricultural production. It is generally believed that China’s agricultural sector has a long-term factor mismatch problem, which not only reduces agricultural total factor productivity, but also leads to a reduction in agricultural output. For example, Yuan and Xie (2011) [34] hold that, if labor misallocation can be effectively eliminated, China’s agricultural total factor productivity can be improved by at least 2%. Chari et al. (2021) [35] pointed out that the elimination of land mismatch could increase China’s land yield and productivity by 8% and 10%, respectively. Ma et al. (2018) [36] proved that human capital mismatch in the agricultural sector reduced agricultural output by 0.61%.

The second is the multi–factor mismatch perspective. Different from the above studies, the existing literature simultaneously brings two or more factors into the analysis framework to demonstrate their impact on agricultural production [37,38]. For example, based on the survey data of micro farmers, Zhu et al. (2011) [39] showed that eliminating the mismatch of capital and labor could increase China’s agricultural total factor productivity by more than 20% under the condition of locking technology. However, limited by the model, they failed to disentangle the separate effects of capital or labor mismatching on agricultural total factor productivity. Chen (2012) [40] pointed out that eliminating cross-sector mismatch of labor, capital, and land could increase China’s agricultural total factor productivity by 6–36%.

The purpose of this paper is to explore the influence of factor mismatch on the high-quality development of agriculture, further analyze the heterogeneity of factor mismatch on high-quality agricultural development, and study its action path, in order to provide an empirical reference for policy makers to reduce the factor mismatch and promote high-quality agricultural development. The above literature has laid a solid theoretical foundation for the development of this paper and provided beneficial inspiration, but there is still room for further improvement. Compared with previous studies, the marginal contribution of this article is mainly reflected in the following three aspects.

Firstly, we consider the substitution effect among factors. Existing literature focuses on the independent impact of single or multiple factors on agricultural production. However, it can be known from economic theory and practice that there exist mutual substitution effects among production factors [41]. Therefore, this paper constructed a mismatch model containing a substitution effect to more accurately describe the factor mismatch coefficient.

Secondly, we reveal the heterogeneity of factor mismatch on high-quality agricultural development. Previous studies mostly focused on the overall effect of factor mismatch on agricultural production [31,39]. This paper further analyzed the effect from the difference in factor allocation modes, geographical locations, and development levels, in order to test whether the influence of factor mismatch on agricultural high-quality development changes with different conditions.

Finally, we explored the mechanism of factor mismatch on high-quality agricultural development. The existing literature of factor mismatch on agricultural production focuses on the sources of factor mismatch [18], but there are relatively few studies concerning the channels through which factor mismatch affects high-quality agricultural development.

The rest of this paper is arranged as follows. Section 2 is the theoretical analysis and hypothesis. Section 3 is the research design. Section 4 is the empirical test, which mainly includes three aspects. Firstly, the status quo of high-quality agricultural development is described. Secondly, the influence of factor mismatch on high-quality agricultural development is tested. Finally, the mechanism of factor mismatch on high-quality agricultural development is revealed. Conclusions and policy recommendations are presented in Section 5.

## 2. Theoretical Analysis and Hypothesis Research

### 2.1. Factor Mismatch and High-Quality Agricultural Development

China’s factor market reform lags behind relatively [16], and the urban–rural dual structure is still prominent, leading to the widespread factor mismatch in China’s agricultural sector, which is highlighted by agricultural labor surplus and insufficient capital input [42]. Factor mismatch not only reduces agricultural total factor productivity, but also restrains the growth of agricultural output. Optimizing factor allocation is the premise to promote high-quality agricultural development [15]. In order to improve the allocation efficiency of production factor, the Third Plenary Session of the 18th CPC Central Committee proposed to let the market play a decisive role in resource allocation, changing the traditional mode of resource allocation in which the government played the leading role and the market played a basic role. Although it has promoted an improvement in factor allocation efficiency, there is still a huge space for factor reallocation in China’s economy [19]. Existing literature shows that, if factor mismatch can be effectively eliminated, China’s agricultural total factor productivity still has room for improvement of 6–36% [40]. On the other hand, under the dual constraints of the fiscal decentralization system and promotion system of local government officials, local protectionism adopted by local governments for economic competition causes serious market segmentation among regions, hinders the free flow of production factor, and leads to significant differences in factor mismatch [39]. Concurrently, owing to differences in natural resource endowment and agricultural development conditions, the level of high-quality agricultural development also shows obvious regional differences [24]. Therefore, the impact of factor mismatch on high-quality agricultural development may vary with the change in external conditions. Based on the above analysis, this paper proposes the following:

**Hypothesis** **1.***The impact of factor mismatch on high-quality agricultural development is heterogeneous*.

### 2.2. Factor Mismatch, Industrial Structure Upgrade, and High-Quality Agricultural Development

Industrial structure upgrading is a key factor in promoting high-quality agricultural development. The optimization and upgrading of the agricultural industrial structure drive the continuous improvement in China’s agricultural green total factor productivity and significantly improve the overall development quality of agriculture [43]. With the improvement in a country’s economic development level, its industrial structure will be constantly be optimized and upgraded with the flow of production factors according to the Petty–Clark theorem. The essence of industrial structure adjustment is the flow and reconfiguration of production elements. However, because of regional market segmentation and factor mismatch caused by local protectionism, agricultural production factors cannot achieve free flow and effective allocation in accordance with market rules, thus inhibiting the optimization and upgrading of industrial structure, leading to the present agricultural industrial structure being far from meeting the requirements of high-quality agricultural development [44]. Cao and Lou (2012) [45] found that factor mismatch would hinder the transformation and upgrading of China’s economy by delaying the change in industrial structure. However, the existing literature does not pay enough attention to the application of this influence channel in agriculture. As China’s economy shifts from high-speed growth to high-quality development, the agricultural industry must keep pace with the economic development stage, otherwise it is difficult to truly achieve high-quality agricultural development. Based on the above analysis, this paper proposes the following:

**Hypothesis** **2.***Factor mismatch reduces the level of high-quality agricultural development by delaying the upgrading of industrial structure*.

### 2.3. Factors Mismatch, Scientific and Technological Progress, and High-Quality Agricultural Development

Under the dual constraints of the fiscal decentralization system and promotion system of local government officials, in order to achieve specific economic goals, local governments will intervene in factor market and place production factors into productive projects with quick results and low risk. Such an intervention will lead to the failure of reasonable allocation of production factors in accordance with the market mechanism, resulting in factor mismatch, and then factor price distortion, which is highlighted by the underestimation of the price of capital, labor, land, and other factors [46], making the price unable to truly reflect the scarcity degree of factors. On the one hand, factor price distortion leads to the emergence of risk-free arbitrage space. Rational innovation subjects tend to seek their own development by competing for tangible factor resources, rather than carrying out high-investment and high-risk technological innovation activities. On the other hand, the underestimation of factor price will inhibit the initiative of innovative talents and is not conducive to the formation and cultivation of human capital, thus squeezing out technological innovation efficiency. Agricultural scientific and technological progress is an important way to realize the transformation from a big agricultural country to an agricultural power. High-quality agricultural development must also be technology-innovation-driven agriculture [47]. However, the situation of insufficient investment in science and technology, which has plagued China’s agricultural development for a long time, has not been fundamentally changed [48], which seriously restricts the improvement in the high-quality agricultural development level. Based on the above analysis, this paper proposes the following:

**Hypothesis** **3:***Factor mismatch inhibits the improvement in high-quality development of agriculture by reducing the innovation level of agricultural science and technology*.

## 3. Model Construction, Variable Description, and Data Sources

### 3.1. Model Construction

This paper constructs the following econometric model to examine the influence of factor mismatch on high-quality agricultural development.
(1)$$Hq_{i,t}=C+\beta_{1}\ln Fm_{i,t}+\beta_{2}X_{i,t}+\mu_{i}+\gamma_{t}+\varepsilon_{i,t}$$

In Formula (1), subscripts *i* and *t* represent province and year, respectively. *C* is the constant term, and *μ_i_* and *γ_t_* represent individual effect and time effect, respectively. *ε_i,t_* is the random interference term, following normal distribution and not correlated with *μ_i_*. *β*_1_ and *β*_2_ are the regression coefficients of the variables. *Hq* is the explained variable, representing the high-quality agricultural development level. *Fm* is the core explanatory variable, representing the factor mismatch of agriculture. *X* is a series of control variables, which will be elaborated in detail below.

### 3.2. Variable Description

#### 3.2.1. Dependent Variable: High-Quality Agricultural Development

High-quality development has rich connotations, so the index system constructed in this paper cannot cover all aspects of it and reflect all of its contents. Considering the availability of data, some indicators have been appropriately abandoned during the construction of the indicator system, although they may have an impact on the high-quality development of agriculture. How to measure high-quality development is the key of this paper; the new development concept (innovation, coordination, green, openness, and sharing) proposed by the Fifth Plenum of the 18th Central Committee Communist Party of China is regarded as an effective criterion for evaluating high-quality development, and is generally recognized by the academic community [49,50]. Therefore, on the basis of an in-depth understanding of new development concepts and combining the existing achievements [22,24,25,49,50], this paper evaluates the high-quality development of agriculture from five dimensions of “innovation level, coordination level, green level, openness level, and sharing level”, and the specific indicators are shown in Table 1.

The entropy method is used to determine the weight coefficient of each index in the evaluation process of high-quality agricultural development. Compared with other methods, the entropy method uses information entropy to measure the variation degree of each index, avoiding the influence of human subjective factors and making the determination of weight more objective and scientific [51]. The calculation steps are as follows:Data standardization processing: In order to eliminate the influence of data dimensional and order of magnitude differences on the calculation results and cause the indicators to have horizontal comparability and applicability, the original data need to be standardized. For positive and negative indicators, the standardized processing method is as follows:

Positive indicators: (2)$$X_{ij}=\frac{x_{ij}-\min\left(x_{j}\right)}{\max\left(x_{j}\right)-\min\left(x_{j}\right)}$$

Negative indicators:(3)$$X_{ij}=\frac{\max\left(x_{j}\right)-x_{ij}}{\max\left(x_{j}\right)-\min\left(x_{j}\right)}$$
where *X_ij_* is the standardized index value; *X_ij_* represents the value of the *j* index in year *i*; and min (*x_j_*) and max (*x_j_*) are the minimum and maximum values of the *j* index, respectively.

2.Calculate the proportion of the *j* index in year *i:*

(4)$$Y_{ij}=x_{ij}/\overset{m}{\underset{i=1}{\sum }}x_{ij}$$
where *m* is the number of years.

3.Calculate the information entropy of the *j* index:


(5)
$$\;e_{j}=-\frac{1}{\ln\left(m\right)}\overset{m}{\underset{i=1}{\sum }}Y_{ij}lnY_{ij}\;0\leq ej\leq 1$$


4.Calculate the redundancy of information entropy:


(6)
$$d_{j}=1-e_{j}$$


5.Calculate the weight of the indicator according to the information entropy redundancy:

(7)$$w_{j}=d_{j}/\overset{n}{\underset{j=1}{\sum }}d_{j}$$
where *n* represents the number of indicators.

6.After the index weight is obtained, the evaluation score of each index can be obtained according to the following formula:

(8)$$S_{ij}=w_{j}X_{ij}$$
where *S_ij_* represents the evaluation score of the *j* index.

7.After obtaining the score of each index, the summary score of high-quality agricultural development can be obtained by summing up according to the following formula:

(9)$$S_{I}=\overset{n}{\underset{j=1}{\sum }}S_{ij}$$
where *S_I_* represents the comprehensive evaluation score of high-quality agricultural development in year *i*.

#### 3.2.2. Independent Variable: Factor Misallocation

Referring to previous research results [52,53], agricultural production factors are divided into capital, labor, and land on the premise that land factor endowment remains unchanged [39]. It is assumed that the production function of each region satisfies the Cobb–Douglas (C–D) form. Based on the premise of profit maximization, Lagrange multiplication is used to calculate the relative mismatch coefficient of capital and labor. The details are as follows:(10)$$\gamma_{Ki}=\left(\frac{K_{i}}{K}\right)/\left(\frac{s_{i}\beta_{Ki}}{\beta_{K}}\right)$$
(11)$$\gamma_{Li}=\left(\frac{L_{i}}{L}\right)/\left(\frac{s_{i}\beta_{Li}}{\beta_{L}}\right)$$

In Formulas (10) and (11), *γ_Ki_* and *γ_Li_* represent the relative mismatch coefficients of capital and labor, respectively. *K_i_* and *L_i_* are the agricultural capital stock and labor stock of region *i*, respectively, whereas *K* and *L* are the agricultural capital stock and labor stock of China, respectively. *S_i_* represents the proportion of agricultural output in region *i* to the whole country. *K_i_*/*K* refers to the actual share of capital usage in region *i* to the whole country; *S_i_β_Ki_*/*β_K_* refers to the theoretical share of capital usage in region *i* to the whole country when capital is effectively allocated. *β_Ki_* and *β_Li_* are capital and labor elasticity in region *i*, respectively. *β_K_* and *β_L_* represent the national capital and labor elasticity, respectively. Based on the C–D production function with constant returns to scale, a variable coefficient panel model with variable intercept and slope is used to estimate the elastic coefficient. The output involved in the calculation process is the gross output value of agriculture, forestry, animal husbandry, and fishery (2004 constant price), and the input is the agricultural capital stock, labor stock, and crop sown area. Agricultural capital stock is measured using the perpetual inventory method and depreciated using the provincial depreciation rate [54]. The labor stock is the number of agricultural laborers in each region multiplied by the average years of education [55]. The sown area of crops is directly based on statistical yearbook data.

The above two formulas take the regional average level as a reference to measure the degree of capital misallocation and labor misallocation, ignoring the incomplete substitution between factors, and failing to reflect whether there is a capital mismatch relative to labor. Therefore, based on the method of Xu and Bai (2017) [56], the factor mismatch coefficient (*Fm*) is further constructed to reflect the degree of capital misallocation relative to labor:(12)$$Fm=\left|\frac{\gamma_{Ki}}{\gamma_{Li}}-1\right|$$

In Formula (4), the larger the *Fm*, the more serious the capital misallocation relative to labor. If *Fm* is zero, there is no factor misallocation.

#### 3.2.3. Other Variables

High-quality agricultural development is not only affected by region factor allocation, but also by other factors. Referring to previous studies, the following control variables were added into the econometric model.

Urbanization level (*Urb*). Modernization cannot be achieved without the coordinated development of urbanization and agricultural modernization. According to the theory of induced technological change, the transfer of the rural population to urban areas makes agricultural labor relatively scarce and induces an improvement in the mechanization level [57]. Then, the improvement in agricultural production efficiency is promoted, contributing to the realization of high-quality agricultural development. This paper adopts the proportion of the urban population at the end of the year to measure the level of urbanization.

Cultivated land quality (*Lq*). Improving the quality of cultivated land is an inevitable requirement for promoting high-quality agricultural development [58]. Agricultural output capacity is an important index to measure the quality of cultivated land, so the yield per unit area of grain is used as an index to measure the quality of cultivated land.

Disaster degree (*Dis*). Natural disasters aggravate the vulnerability of agricultural production. This paper uses the ratio of the area that has become a disaster to the area that has been affected by natural disasters to represent the degree of agricultural disaster. The larger the value, the greater the impact on the high-quality development of agriculture. The area that has become a disaster reflects the ability of each region to cope with natural disasters, and the area affected by disasters represents the impact of uncontrollable climatic factors on agricultural production.

Energy consumption (*Ec*). Energy consumption is the main source of agricultural carbon emissions [59]. Although it provides dynamic support for the high-quality development of agriculture, it brings a lot of agricultural pollution because of its low utilization efficiency. This paper chooses per capita energy consumption as a proxy index of energy consumption.

Industrial level (*Ind*). The improvement in the level of industrialization promotes the rapid development of petroleum agriculture and reduces the agricultural ecological efficiency [60], which has an impact on the high-quality development of agriculture. This paper adopts the proportion of the output value of the secondary industry to the regional GDP for measurement.

Economic development level (*Pgdp*). The economic development level is closely related to residents’ income, and the expansion of the regional economic scale will promote the improvement in residents’ income level. In recent years, China’s per capita GDP has been increasing. According to Maslow’s hierarchy of needs theory, residents’ demand for high-quality agricultural products increases with their income. This paper adopts per capita GDP to measure the regional economic development level [61].

Financial support for agriculture (*Sup*). Fiscal expenditure in agriculture is conducive to improving agricultural production conditions, promoting agricultural scientific and technological innovation, and improving agricultural productivity [62]. This paper uses the ratio of regional financial expenditure to total financial expenditure as a proxy index.

Soil and water conservation (*Wl*). Water and soil resource protection is the core of agricultural development [63]. The soil erosion control area is selected to represent the text.

#### 3.2.4. Intermediary Variables

Industrial structure upgrade (*Ts*). Industrial structure changes with an improvement in the economic development level. The adjustment of agricultural industrial structure improves the efficiency of resource allocation and contributes to the promotion of high-quality agricultural development. According to the Petty–Clark theorem, this paper uses the proportion of agriculture, forestry, animal husbandry, and fishery services to measure the industrial structure upgrade.

Agricultural science and technology progress (*Tech*). Technological progress is an important factor to promote the high-quality development of agriculture. In recent years, the contribution rate of technological progress to China’s agricultural development has reached over 50% [64]. Referring to previous research results [65], the contribution rate of agricultural scientific and technological progress was used as a representation index of agricultural scientific and technological progress.

### 3.3. Data Sources

In this paper, 527 samples from 31 provinces in China (except Hong Kong, Macao, and Taiwan) from 2004 to 2020 were selected for analysis. The reason for taking 2004 as the starting point is that the Chinese government issued the first policy document supporting agricultural development in the 21st century in that year. It was also the starting point of China’s 19th consecutive increase in grain output, laying a solid foundation for the high-quality development of agriculture. All data in this paper are from China statistical yearbook (2005–2021), provincial statistical Yearbook and statistical bulletin of National Economic and Social Development. A few missing values are complemented by the linear interpolation method. Descriptive statistics of all variables are shown in Table 2. It can be seen that the mean values of most variables are greater than the standard deviation, indicating that the degree of data dispersion is not high, which can be analyzed in the next step.

## 4. Results’ Analysis

### 4.1. Dynamic Evolution Characteristics of High-Quality Agricultural Development

Figure 1 depicts the evolution characteristics of high-quality agricultural development in China using the kernel density map. By shifting the position of the curve to the left and right to reflect the level of high-quality agricultural development, curve kurtosis represents the divergence or polarization trend of high-quality agricultural development. The curve shape indicates the degree of convergence or diffusion of high-quality agricultural development. Firstly, the overall position of the curve fluctuates to the right from 2004 to 2020, indicating that the high-quality development of agriculture is constantly improving, but the development level of several observation periods after the sample years has declined. Secondly, the peak height decreased year by year from peak to wide peak, and the coverage area of the curve increased, indicating that the spatial difference of high-quality agricultural development in the sample period was expanding. Thirdly, the trailing of the curve fluctuates and shrinks on the right side, and the ductility of the distribution has a shrinking trend to some extent, indicating that the difference between provinces in the high-value region and the average level of high-quality agricultural development has narrowed. Finally, kernel density was experienced from the twin peaks to unimodal curve evolution, of which 2004 bimodal distribution is significant, but the height of the peak between the main and side gap is larger, and the year distribution curve of the left and right sides of the ductility reach the extreme value of all years, meaning that the polarization characteristics of high-quality agricultural development, but then the year is given priority with unimodal, it shows that the high-quality development of agriculture is changing from diffusion to convergence, and the polarization phenomenon is weakening. In conclusion, although the level of high-quality agricultural development is improving, the development level in most provinces is still low, and the spatial disparity is expanding.

### 4.2. Benchmark Regression

In this paper, we use a bidirectional fixed effect model that controls individual and time to estimate the benchmark model, and take the logarithm to eliminate the effect of heteroscedasticity for all absolute data. As this paper uses short panel data, the generalized least squares method cannot be used to solve the autocorrelation and cross-sectional correlation problems in the model. Therefore, the Driscol–Kraay standard deviation processing model, which can solve heteroscedasticity, autocorrelation, and cross-sectional correlation simultaneously, and is suitable for short panel data, is adopted. Table 3 reports the full-sample baseline regression results for the impact of factor mismatches on high-quality agricultural development. Column (1) is the regression result without the addition of control variables, which tests the direct impact of factor mismatch on the high-quality development of agriculture. The result shows that factor mismatch is significantly negative at the 1% level. Columns (2) to (5) are the regression results after gradually adding control variables. We found that the coefficient of factor mismatch was always significantly negative and passed the 1% significance level test; only the coefficient was different, indicating that factor mismatch could significantly inhibit the improvement in the high-quality agricultural development level. The research results, to some extent, support the theoretical analysis of Huang (2021) [47].

We use the results in column (5) for analysis, and the economic meaning of the factor mismatch regression coefficient of –0.026 is as follows: when other conditions remain unchanged, the factor mismatch degree increases by 1%, and the high-quality agricultural development score decreases by an average of 0.026% units. In other words, keeping other conditions unchanged, there is a difference of 0.026% units in the high-quality agricultural development score between the extremely distorted factor allocation and the extremely reasonable factor allocation. The average value of high-quality agricultural development is 0.33, and because of extreme differences in factor allocation, there will be a difference of 0.01% (0.33 × 0.026%) units in the score of high-quality agricultural development.

In terms of control variables, the regression coefficient of the urbanization level is significantly positive, indicating that the improvement in urbanization level has a positive impact on the high-quality development of agriculture. The rapid improvement in China’s urbanization level can lay a foundation for large-scale agricultural operation, improve the allocation of agricultural resources, and then promote the improvement of high-quality development of agriculture [66,67,68]. The regression coefficient of cultivated land quality is significantly negative. Long-term unreasonable land use has led to the continuous decline in cultivated land quality, hindering the high-quality development of agriculture [69,70]. The coefficient of the disaster rate is significantly negative, indicating that the occurrence of natural disasters in agriculture is not conducive to the high-quality development of agriculture. Per capita energy consumption has a significant negative effect on the high-quality development of agriculture, because the increase in energy consumption will reduce the green total factor productivity of agriculture, thereby inhibiting the steady improvement in high-quality agricultural development. The coordinated development level of industrialization and agricultural modernization in China is relatively low [71], which significantly inhibits the improvement in high-quality agricultural development. The coefficient of economic development level is significantly positive, and the enhancement of regional economic strength will help increase investment in agricultural science and technology and promote the high-quality development of agriculture. The coefficient on the level of fiscal support for agriculture is negative, but not significant. As food security has always been China’s strategic goal, the investment of fiscal support for agriculture has focused too much on the growth in agricultural quantity and neglected the improvement in quality [72]. The regression coefficient of soil and water conservation is positive, but not significant, indicating that increasing the protection of soil and water resources can effectively promote the improvement in the high-quality agricultural development level, but the current soil and water loss control area accounts for a small proportion of agricultural sown area, which fails to give full play to its promoting role.

### 4.3. Robustness and Endogenous Check

In order to ensure the robustness and reliability of the empirical results mentioned above, this paper mainly conducted robustness tests from changing the sample size, adjusting the sample period, and endogenous treatment, as shown in Table 4.

(1)Changing the sample size. In order to avoid the influence of extreme values on the regression results, 1% bilateral tail reduction was applied to all variables and the removed samples were re-estimated. The results are shown in Column (1) of Table 4. After the removal of extreme values, factor mismatch still has a significant inhibitory effect on high-quality agricultural development, which is consistent with the benchmark regression result.(2)Adjusting the sample period. In 2006, the Agricultural Tax Regulations of the People’s Republic of China was formally abolished, which reduced the cost of agricultural production and operation and improved the international competitiveness of agricultural products [73]. To some extent, it has changed the external environment faced by agricultural development. Based on this, this paper selected samples after 2006 for re-regression; the results in Column (2) of Table 4 show that the regression coefficient of factor mismatch is significantly negative, which further indicates that the conclusion that factor mismatch inhibits high-quality agricultural development is robust.(3)In order to avoid the model setting error and endogeneity problem of missing variables, this paper introduces the lagging term of explained variables. Two-step differential GMM and two-step system GMM estimation methods are used to deal with the endogeneity problems caused by the introduction of lag terms of explained variables. The results in columns (3) and (4) of Table 4 show that the residual term of the equation has first-order sequence correlation, but no second-order sequence correlation, indicating that the model’s set is reasonable. The Hansen overidentification test accepts the null hypothesis that tool variables are effective, indicating that the choice of tool variables is reasonable. After considering the dynamic effect of the model, the impact of factor mismatch on high-quality agricultural development still remains negative and significant. Moreover, high-quality agricultural development has significant “time inertia” and has certain path-dependent characteristics, that is, if the level of agricultural development is high in the current period, it may continue to be high in the next period, showing a certain “snowball effect”. At the same time, all explanatory variables lag one period to effectively avoid possible reverse causality. The results in Column (5) of Table 4 show that factor mismatch is significantly negative at 1% level, which is consistent with the benchmark regression results.

The results of several robustness tests show that the model has good explanatory power, and the direction of the core explanatory variable coefficients has good consistency; only the significance has changed, indicating that the main research conclusions of this paper have sound robustness.

### 4.4. Heterogeneity Analysis

The conclusions above indicate that there are significant regional differences in high-quality agricultural development, and regional market segmentation leads to different agricultural factors mismatch [18]. Therefore, only considering the average effect of national factor mismatch on agricultural high-quality development may lead to distortion and misjudgment. In view of this, the samples were grouped according to the difference in high-quality development level, geographical location heterogeneity, and factor allocation mode to test the robustness of the baseline regression results. The estimated results are shown in columns (1) to (7) of Table 5.

(1)Heterogeneity of development level. According to previous literature, optimizing factor allocation is the prerequisite for high-quality agricultural development [15]. Therefore, areas with possibly high development level are more affected by factor mismatch. Hence, according to the annual average of the high-quality agricultural development score, the study area is divided into two categories: high-level and low-level. If the score of high-quality agricultural development in an area exceeds the annual average, it is considered as a high-level area, otherwise it is considered as a low-level area. For details, see columns (1) and (2) in Table 5. The results show that, regardless of the level of development, factor mismatch has a significant negative impact on high-quality agricultural development, but it has a greater impact on high-quality agricultural development in high-level areas than in low-level areas.(2)Geographical location heterogeneity. The previous literature generally believed that there were regional differences in high-quality agricultural development [24]. So, does the impact of factor mismatch on high-quality agricultural development vary with different regions? Therefore, this article divides 31 provinces in China into three regions: eastern, central, and western, for verification, as shown in columns (3) to (5) in Table 5. The results show that factor mismatch has a negative impact on high-quality agricultural development in different regions, but the impact on the western region is not significant. The reason may be that the factor mismatch degree in western China is relatively light, and the high-quality agricultural development in this region is mainly restricted by the regional natural resource endowment and economic development level [74], leading to the weakening of the marginal effect of factor mismatch.(3)Heterogeneity of the factor allocation mode. In 2013, the Third Plenary Session of the 18th Central Committee of China proposed to allow the market to play a decisive role in resource allocation. This means that the allocation of factors is dominated by the government to give way to the market. The market-based allocation of factors improves the efficiency of resource allocation and alleviates the degree of factor mismatch, which is an effective way to promote high-quality economic development [11]. This paper divides the samples into two groups around 2013, and tests the relationship between factor mismatch and agricultural high-quality development before and after the market-oriented factor allocation. The specific regression results are shown in columns (6) and (7) of Table 5. The results show that factor mismatch has a significant inhibiting effect on the improvement in the high-quality agricultural development level before market allocation. After the market-oriented allocation, although the regression coefficient of factor mismatch does not have statistical significance, the absolute value of its coefficient decreases significantly, which indicates, to some extent, that the change in factor allocation mode improves the efficiency of resource allocation and alleviates the negative impact of factor mismatch on high-quality agricultural development.

Based on the above analysis, it can be seen that the impact of factor mismatch on high-quality agricultural development is different with the level of development quality, the change in geographical location, and the different modes of factor allocation. Thus, hypothesis 1 was verified.

### 4.5. Mechanism Test

Based on the above theoretical analysis, this paper holds that factor mismatch hinders the improvement in high-quality agricultural development by delaying the upgrading of industrial structure and reducing the level of agricultural science and technology. In order to effectively identify this transmission mechanism, the following recursive model is constructed to examine how industrial structure upgrading and agricultural science and technology progress play a role in the process of factor mismatch affecting high-quality agricultural development by referring to the mediation effect test method proposed by Baron and Kenny (1986) [75] and Wen and Ye (2014) [76].
(13)$$Hq_{i,t}=C+\beta_{1}\ln Fm_{i,t}+\beta_{2}X_{i,t}+\mu_{i}+\gamma_{t}+\varepsilon_{i,t}$$
(14)$$Med_{i,t}=C+\theta_{1}\ln Fm_{i,t}+\theta_{2}X_{i,t}+\mu_{i}+\gamma_{t}+\varepsilon_{i,t}$$
(15)$$Hq_{i,t}=C+\varphi_{1}\ln Fm_{i,t}+\varphi_{2}Med_{i,t}+\varphi_{3}X_{i,t}+\mu_{i}+\gamma_{t}+\varepsilon_{i,t}$$

Firstly, the benchmark model (13) was regressed to test the relationship between factor mismatch and high-quality agricultural development. If the *β*_1_ regression coefficient is significant, it means that, during the sample period, factor mismatch will have an impact on the high-quality development of agriculture, and if it is not significant, the test is stopped. Secondly, regression Equation (14) was performed to test the relationship between factor mismatch and mediating variables. If the *θ*_2_ regression coefficient is significant, it means that factor mismatch will have an impact on the mediating variable. Finally, regression is performed on Formula (15); if the coefficients of factor mismatch and mediating variable are both significant, and the coefficient *φ*_1_ < *β*_1_, this means that there is a partial mediation effect. If the coefficient *φ*_1_ is not significant, but the coefficient *φ*_2_ is significant, this means that there is a complete mediation effect.

Table 6 reports the test results of the mechanism of industrial structure upgrading and high-quality agricultural development. Among them, column (1) is the estimation result of the benchmark model (1), so it is the same as the regression result in column (5) of Table 3. Columns (2) and (3) are the estimated results of Formula (14) with industrial structure upgrading and agricultural science and technology progress as explained variables, respectively. From the results, we found that the regression coefficients of factor mismatch were all significantly negative, indicating that factor mismatches delayed the upgrading of the industrial structure and lowered the level of agricultural science and technology progress. On the one hand, factor mismatch prevents the rational flow and allocation of production factors in accordance with market rules, hindering the optimization and upgrading of the industrial structure. On the other hand, the arbitrage space formed by factor mismatch stimulates innovative subjects to use tangible factors and reduces the enthusiasm of innovative talents to carry out innovation activities. In turn, the factor mismatch has a negative effect on the comprehensive effect of industrial structure upgrading and agricultural science and technology progress.

Columns (4–6) further report the regression of factor mismatch and intermediary variables on high-quality agricultural development, that is, the results of Formula (15). The results show that, as expected, both mediating variables contributed to the improvement in the high-quality agricultural development level. In addition, compared with the benchmark regression results in column (1), after adding the intermediary variables industrial structure upgrade (column (4)) and agricultural science and technology progress (column (5)), respectively, the estimated coefficient of factor mismatch on high-quality agricultural development has declined to a certain extent, which preliminarily shows the existence of the intermediary effect of industrial structure upgrading and agricultural scientific and technological progress. Further, after adding the intermediary variables’ industrial structure upgrade and agricultural science and technology progress at the same time (column (6)); although the estimated coefficient of factor mismatch is not statistically significant, the value has further decreased, which means that the upgrading of industrial structure and the advancement in agricultural science and technology are important mechanisms for factor mismatch to inhibit the improvement in high-quality agricultural development. Therefore, hypotheses 2 and 3 are validated.

## 5. Conclusions

Optimizing factor allocation is the premise of promoting high-quality development of agriculture. Based on the panel data of 31 provinces in China from 2004 to 2020, this paper analyzes the impact of factor mismatch on high-quality agricultural development. The results show the following: (1) The level of high-quality agricultural development is constantly improving, but the development level of most provinces is still low, and the inter-provincial differences are expanding. (2) Factor mismatch has a significant negative impact on the level of high-quality agricultural development, and this effect has temporal and spatial heterogeneity. From the perspective of time heterogeneity, from 2004 to 2013, factor mismatch had a significant negative impact on high-quality agricultural development. However, from 2014 to 2020, as the market played a decisive role in resource allocation, the effect of factor mismatching on high-quality agricultural development was negative, but not significant. From the perspective of spatial heterogeneity, factor mismatch has a significant negative effect on high-quality agricultural development in eastern and central China, but not in western China. In addition, this paper also found that, the higher the level of high-quality development, the greater the impact of factor mismatch on it. (3) The mediating effect of mismatch factors on high-quality agricultural development is significant. With the optimization and upgrading of industrial structure and the improvement in the agricultural science and technology level, the negative impact of mismatched factors on high-quality agricultural development will be constantly weakened.

Based on the above research conclusions, this paper argues that (1) In view of the fact that the inter-provincial differences in high-quality agricultural development are expanding, provinces should formulate agricultural development policies according to their resource endowment and the reality of agricultural development. For regions with a high level of agricultural development, a replicable and popularized agricultural development model should be explored to provide a reference for agricultural development in other regions on the basis of maintaining the existing level of development. For regions with relatively low levels of agricultural development, it is necessary to learn the agricultural development models of regions with high levels of development on the basis of fully exploiting the endogenous power of local agricultural development, and combine the actual conditions of regional agricultural development to summarize a specific model for the high-quality development of local agriculture. (2) According to the temporal and spatial heterogeneity of factors mismatch affecting high-quality agricultural development. In areas with relatively high levels of agricultural development, such as the eastern and central regions, market segmentation should be further eliminated and free flow and optimal allocation of factors of production should be promoted in accordance with China’s specific opinions on market-oriented reform of factors of production and the actual conditions of the region. (3) Provinces should further strengthen their support for agricultural science and technology and improve the process of agricultural science and technology and promotion. At the same time, on the basis of ensuring national food security, combined with local agricultural development conditions, the optimization and upgrading of agricultural industrial structure should be further promoted, full play should be given to the weakening effect of agricultural scientific and technological progress and industrial structure adjustment on factor mismatch, and then the improvement in the high-quality agricultural development level should be promoted.

The limitation of this manuscript lies in the lack of statistical data, which makes it impossible for us to conduct an in-depth analysis of the research topic from the prefecture-level city or smaller spatial units. However, the analysis based on the provincial level does not affect the generality of the conclusions. In order for our research conclusions to have stronger policy reference value in future research, we plan to narrow the research scope to the main agricultural planting provinces in China or the main grain-producing areas in northeast China, so that we can obtain various data required for the research through field research, so as to more accurately analyze the impact of factor mismatch on high-quality agricultural development.

## Figures and Tables

**Figure 1 ijerph-19-09804-f001:**
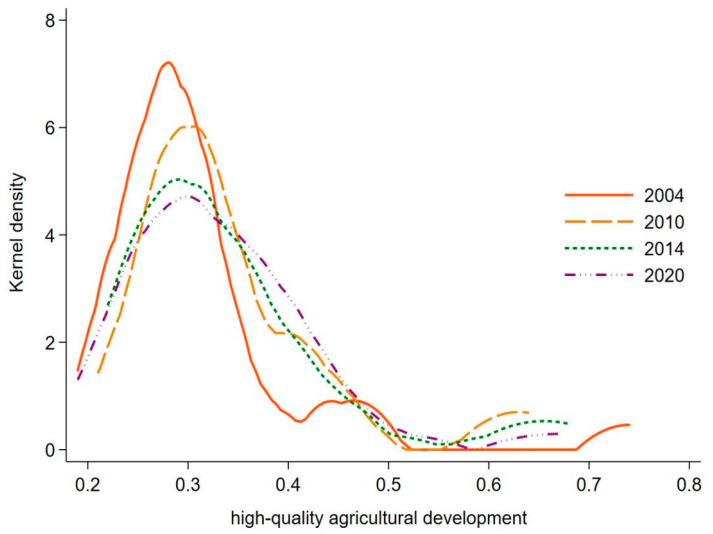
Kernel density of high-quality agricultural development from 2004 to 2020.

**Table 1 ijerph-19-09804-t001:** Evaluation the index system of high-quality agricultural development.

Primary Indexes	Secondary Indexes	Tertiary Indexes (Property)
Innovation level	Innovation base	A1: Proportion of agricultural science and technology personnel (+)
		A2: Proportion of investment in agricultural research (+)
		A3: Proportion of agricultural science and technology institutions (+)
	Innovation output	A4: Proportion of agricultural patents granted (+)
		A5: Agricultural labor productivity (+)
		A6: Agricultural land productivity (+)
Coordination level	Industrial coordination	B1: Proportion of rural non–farm employment (+)
		B2: Industrial structure adjustment index (+)
	Urban and rural coordination	B3: Binary contrast coefficient (+)
Green level	Resource utilization	C1: Water–saving irrigation area intensity (+)
		C2: Fertilizer utilization intensity (–)
		C3: Pesticide utilization intensity (–)
		C4: Agricultural film utilization intensity (–)
		C5: Agricultural water consumption (–)
		C6: Agricultural electricity consumption (–)
	Environmental impact	C7: Agricultural carbon intensity (–)
		C8: Agricultural non-point source pollution (–)
Open level	Foreign trade	D1: Foreign trade dependence of agricultural products (+)
	Foreign investment	D2: Intensity of foreign investment utilization in agriculture (+)
Sharing level	Welfare sharing	E1: Education level (+)
		E2: Public health level (+)
	Benefit sharing	E3: Urban–rural income ratio (–)
		E4: Urban–rural consumption ratio (–)
		E5: Engel coefficient (–)

Note: the detailed explanation of the tertiary indicators is placed in Appendix A in Appendix A.

**Table 2 ijerph-19-09804-t002:** Descriptive statistics.

Variable Types	Variable Name	Obs.	Mean	S.D.	Min	Max
Dependent variable	High-quality agricultural development	527	0.33	0.10	0.18	0.74
Independent variable	factor misallocation	527	1.59	2.31	0.00	12.91
Intermediary variable	Industrial structure upgrade	527	0.04	0.02	0.01	0.11
	Agricultural science and technology progress	527	0.32	0.14	0.07	0.78
Other variable	Urbanization level	527	0.52	0.16	0.16	0.90
	Cultivated land quality	527	5561.67	945.36	3214.76	8214.00
	Disaster degree	527	0.48	0.16	0.00	0.90
	Energy consumption	527	3.37	1.76	0.86	11.00
	Industrial level	527	0.45	0.09	0.16	0.66
	Economic development level	527	31,307.35	19,448.86	4317.00	117,139.00
	Financial support for agriculture	527	0.10	0.03	0.02	0.20
	Soil and water conservation	527	3565.67	2915.58	15.22	14,625.00

**Table 3 ijerph-19-09804-t003:** Benchmark regression results.

Variable	(1)	(2)	(3)	(4)	(5)
*Fm*	–0.029 ***	–0.024 ***	–0.026 ***	–0.026 ***	–0.026 ***
	(−3.949)	(−2.690)	(−3.364)	(−3.068)	(−3.303)
*Urb*		0.109 *	0.132 *	0.109 *	0.109 *
		(1.735)	(2.010)	(1.784)	(1.787)
*Lq*		–0.051 ***	–0.055 ***	–0.053 ***	–0.046 ***
		(−2.870)	(−3.188)	(−3.033)	(−2.919)
*Dis*			–0.014 *	–0.015 **	–0.016 **
			(−2.021)	(−2.134)	(−2.106)
*Ec*			–0.028 ***	–0.028 ***	–0.030 ***
			(−2.848)	(−3.327)	(−3.249)
*Ind*				–0.075 *	–0.088 ***
				(−1.967)	(−2.959)
*Pgdp*				0.030 **	0.029 **
				(2.321)	(2.343)
*Sup*					–0.087
					(−1.123)
*Wl*					0.004
					(1.452)
*C*	0.324 ***	0.723 ***	0.772 ***	0.520 ***	0.447 **
	(77.424)	(5.194)	(5.929)	(2.730)	(2.541)
*Province*	YES	YES	YES	YES	YES
*Year*	YES	YES	YES	YES	YES
R^2^	0.3021	0.3309	0.3512	0.3601	0.3647
*N*	527	527	527	527	527

Note: * *p* < 0.1, ** *p* < 0.05, *** *p* < 0.01, *t* values in parentheses.

**Table 4 ijerph-19-09804-t004:** Robustness check results.

Variable	Robustness Check		Endogenous Check		
	(1)	(2)	(3)	(4)	(5)
*L.Hq*			0.460 **	0.978 ***	
			(2.192)	(23.564)	
*Fm*	−0.025 ***	−0.015 ***	–0.019 *	–0.004 **	–0.015 ***
	(−3.156)	(−3.246)	(–1.708)	(–2.386)	(–3.226)
*Urb*	0.094	0.353 ***	0.367 ***	0.035	0.183 ***
	(1.541)	(6.202)	(2.772)	(1.154)	(5.551)
*Lq*	−0.039 **	−0.044 ***	–0.007	–0.001	–0.064 ***
	(−2.714)	(−3.081)	(–0.221)	(–0.071)	(–3.332)
*Dis*	−0.015 **	−0.016 ***	–0.003	–0.002	–0.019 ***
	(−2.085)	(−3.564)	(–0.577)	(–0.299)	(–2.645)
*Ec*	−0.041 ***	−0.037 ***	0.008	–0.001	–0.023 **
	(−5.149)	(−3.131)	(0.384)	(–0.185)	(–2.012)
*Ind*	−0.091 **	−0.017	–0.001	0.018	–0.108 ***
	(−2.516)	(−0.855)	(–0.010)	(0.635)	(–3.072)
*Pgdp*	0.036 ***	0.009	–0.037 *	–0.006	0.038 ***
	(2.751)	(0.901)	(–1.761)	(–0.271)	(2.834)
*Sup*	−0.099	−0.173 ***	0.095	0.008	–0.041
	(−1.390)	(−3.951)	(1.134)	(0.089)	(–0.599)
*Wl*	0.007 **	0.005 *	–0.004	–0.001	0.006 *
	(2.377)	(1.721)	(–0.850)	(–0.733)	(1.778)
*C*	0.311 *	0.268 *			0.479 **
	(1.787)	(1.832)			(2.286)
*Province*	YES	YES	YES	YES	YES
*Year*	YES	YES	YES	YES	YES
AR(1)			0.036	0.010	
AR(2)			0.903	0.949	
Hansen			0.694	0.897	
R^2^	0.3723	0.2944			0.285
*N*	527	465	434	465	496

Note: * *p* < 0.1, ** *p* < 0.05, *** *p* < 0.01, *t* values in parentheses.

**Table 5 ijerph-19-09804-t005:** Heterogeneity test results.

Variable	Development Level		Geographical Location			Factor Allocation mode	
	Low-Level	High-Level	East	Central	West	Government	Market
	(1)	(2)	(3)	(4)	(5)	(6)	(7)
*Fm*	−0.018 **	−0.034 ***	−0.030 **	−0.105 ***	−0.007	−0.028 ***	−0.012
	(−2.559)	(−4.718)	(−2.415)	(−7.271)	(−0.888)	(−3.395)	(−0.878)
*Urb*	0.076 **	0.169 ***	0.143	0.003	−0.005	0.033	0.540 ***
	(2.561)	(3.008)	(1.587)	(0.047)	(−0.229)	(0.956)	(7.980)
*Lq*	−0.018	−0.100 **	−0.144 **	0.039	−0.023 *	−0.003	−0.061 ***
	(−1.456)	(−2.506)	(−2.985)	(1.161)	(−2.108)	(−0.105)	(−4.660)
*Dis*	−0.015 *	−0.011	−0.019 **	0.001	−0.033 **	−0.034 ***	−0.015 ***
	(−1.756)	(−0.948)	(−2.928)	(0.194)	(−2.967)	(−2.806)	(−6.735)
*Ec*	−0.020 **	−0.003	0.031	−0.018	−0.030 **	0.002	0.027 **
	(−2.243)	(−0.118)	(1.044)	(−0.733)	(−2.396)	(0.168)	(2.088)
*Ind*	0.006	−0.219 ***	−0.109	0.017	−0.002	−0.145 ***	0.063 *
	(0.177)	(−3.369)	(−0.639)	(0.544)	(−0.057)	(−3.827)	(1.900)
*Pgdp*	0.042 ***	0.009	−0.028	0.008	0.038 ***	−0.037 ***	−0.026 **
	(5.630)	(0.338)	(−0.935)	(0.296)	(3.395)	(−2.804)	(−2.351)
*Sup*	−0.116	−0.008	−0.092	−0.176	−0.041	0.021	−0.192 ***
	(−1.116)	(−0.054)	(−0.692)	(−1.120)	(−0.301)	(0.298)	(−3.477)
*Wl*	−0.002	0.032 ***	0.028 ***	0.006	0.002	−0.005	0.037 ***
	(−0.683)	(3.832)	(3.350)	(1.025)	(0.686)	(−0.783)	(6.019)
*C*	0.047	0.982 **	1.730 ***	−0.140	0.129	0.795 **	0.467
	(0.348)	(2.244)	(4.254)	(−0.303)	(0.898)	(2.723)	(1.534)
*Province*	YES	YES	YES	YES	YES	YES	YES
*Year*	YES	YES	YES	YES	YES	YES	YES
R^2^	0.5678	0.434	0.5326	0.7090	0.4374	0.4620	0.4103
*N*	200	327	187	136	204	279	248

Note: * *p* < 0.1, ** *p* < 0.05, *** *p* < 0.01, *t* values in parentheses.

**Table 6 ijerph-19-09804-t006:** Mechanism test results.

Variable	*Hq*	*Ts*	*Tech*	*Hq*	*Hq*	*Hq*
	(1)	(2)	(3)	(4)	(5)	(6)
*Fm*	–0.026 ***	−0.015 ***	−0.048 ***	−0.020 ***	−0.013 **	−0.006
	(−3.303)	(−4.606]	(−3.771)	(−3.468)	(−2.127)	(−1.141)
*Ts*				0.410 ***		0.474 ***
				(5.015)		(10.728)
*Tech*					0.272 ***	0.276 ***
					(10.945)	(12.756)
*Urb*	0.109 *	−0.011 *	0.354 ***	0.114 *	0.013	0.016
	(1.787)	(−1.741)	(3.477)	(1.906)	(0.307)	(0.420)
*Lq*	–0.046 ***	0.005	−0.061 *	−0.048 ***	−0.03	−0.032
	(−2.919)	(0.570)	(−1.796)	(−3.064)	(−1.451)	(−1.685)
*Dis*	–0.016 **	−0.005 *	−0.033 *	−0.014 *	−0.007	−0.004
	(−2.106)	(−1.871)	(−1.744)	(−2.034)	(−1.295)	(−0.912)
*Ec*	–0.030 ***	−0.006	0.039	−0.028 ***	−0.041 ***	−0.038 ***
	(−3.249)	(−1.268)	(1.060)	(−3.126)	(−3.702)	(−3.689)
*Ind*	–0.088 ***	0.015	−0.477 ***	−0.094 ***	0.041 **	0.036 **
	(−2.959)	(0.998)	(−7.063)	(−3.127)	(2.488)	(2.658)
*Pgdp*	0.029 **	0.009	0.075 ***	0.025 **	0.009	0.004
	(2.343)	(0.943)	(3.436)	(2.178)	(0.962)	(0.496)
*Sup*	–0.087	0.017	−0.074	−0.094	−0.067	−0.075
	(−1.123)	(0.318)	(−0.269)	(−1.323)	(−1.403)	(−1.674)
*Wl*	0.004	−0.006 ***	0.01	0.007 **	0.002	0.004 **
	(1.452)	(−3.280)	(1.506)	(2.748)	(0.750)	(2.720)
*C*	0.447 **	−0.054	0.143	0.469 ***	0.408 *	0.433 **
	(2.541)	(−0.341)	(0.424)	(2.795)	(1.809)	(2.212)
*Province*	YES	YES	YES	YES	YES	YES
*Year*	YES	YES	YES	YES	YES	YES
R^2^	0.3647	0.3432	0.4363	0.3886	0.6019	0.6336
*N*	527	527	527	527	527	527

Note: * *p* < 0.1, ** *p* < 0.05, *** *p* < 0.01, *t* values in parentheses.

## Data Availability

All data in the manuscript are from the statistical yearbook compiled by the National Bureau of Statistics of China (http://www.stats.gov.cn/, accessed on 15 February 2022). If the reader needs the data in the manuscript for academic research, please contact the author via email.

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
