# Peer review of "High-Quality Development of Chinese Agriculture under Factor Misallocation"

_ijerph, 2022, doi:10.3390/ijerph19169804_

Round 1
Reviewer 1 Report
The article is interesting and focused on important issues of agricultural development. Paper structure is correct. Introduction describes clearly major problems of agricultural development. The main purpose of the paper and hypotheses are clearly described. The article has a very good literature background. Data used in the research is current, however I would recommend describing more detailed how the sample was selected, f.e. what criteria were used. Research results are supported by literature and have recommendations for practice. In conclusion part I would recommend describing the limitations of the study and some directions for future studies.
Author Response
Dear reviewers, on behalf of all authors, thank you for your affirmation of our manuscript, we have carefully revised it according to your valuable comments, please check the attachment

Reviewer 2 Report
The paper is disappointing. It does not provide the methods employed to derive the data used in the analysis. Moreover, the sample is too small to do a meaningful analysis let alone draw policy conclusions. The authors cite numerous articles that employ similar approaches with a focus on economic development in China, However, those papers use larger data sets or if they use a similarly small sample focus on a very well-defined question that can be answered with the data. (see attached file for some more detailed remarks). Note, I have not done a serious check on English grammar, syntax an style.
Because most of the methods employed are not described in detail, there may be other serious flaws in the paper that I cannot currently determine. The methods and results presented in the manuscript leave much to be desired.

Author Response
Dear reviewers, on behalf of all authors, thank you for your careful review of our manuscript, your suggestions have brought great help to the improvement of the quality of our manuscript, we have made revisions according to your suggestions, please check the attachment

Reviewer 3 Report
It's a good job. It has some punctuation and formatting errors. Review the nomenclature of Tables 3, 4, 5 and 6.
Author Response
Dear reviewers, on behalf of all authors, thank you for your affirmation of our manuscript, we have carefully revised your suggestions, please check the attachment

Round 2
Reviewer 2 Report
Unfortunately the authors have not adequately addressed any of the major concerns. Only the minor issues that required little or no effort were addressed.
The authors have added a statement about the limitations of their study in the conclusions section but then state that the limitations have no effect on the generalized conclusions they draw. I completely disagree. See my previous comments.
Author Response
Dear anonymous review expert,
We have made supplementary explanations for some questions. Thank you for your valuable comments on our manuscript. Please check the attachment for details.
